# Flavones, Flavonols, Lignans, and Caffeic Acid Derivatives from *Dracocephalum moldavica* and Their In Vitro Effects on Multiple Myeloma and Acute Myeloid Leukemia

**DOI:** 10.3390/ijms232214219

**Published:** 2022-11-17

**Authors:** Karin Jöhrer, Mayra Galarza Pérez, Brigitte Kircher, Serhat Sezai Çiçek

**Affiliations:** 1Tyrolean Cancer Research Institute, Innrain 66, 6020 Innsbruck, Austria; 2Department of Pharmaceutical Biology, Kiel University, Gutenbergstraße 76, 24118 Kiel, Germany; 3Department of Internal Medicine V (Hematology and Oncology), Medical University Innsbruck, Anichstraße 35, 6020 Innsbruck, Austria

**Keywords:** hematologic cancer, FLT3 inhibitor, natural product, cytotoxicity, apoptosis, structure-activity relationship, apigenin, luteolin, quercetin, kaempferol

## Abstract

Phenolic plant constituents are well known for their health-promoting and cancer chemopreventive properties, and products containing such constituents are therefore readily consumed. In the present work, we isolated 13 phenolic constituents of four different compound classes from the aerial parts of the Moldavian dragonhead, an aromatic and medicinal plant with a high diversity on secondary metabolites. All compounds were tested for their apoptotic effect on myeloma (KMS-12-PE) and AML (Molm-13) cells, with the highest activity observed for the flavone and flavonol derivatives. While diosmetin (**6**) exhibited the most pronounced effects on the myeloma cell line, two polymethylated flavones, namely cirsimaritin (**1**) and xanthomicrol (**3**), were particularly active against AML cells and therefore subsequently investigated for their antiproliferative effects at lower concentrations. At a concentration of 2.5 µM, cirsimaritin (**1**) reduced proliferation of Molm-13 cells by 72% while xanthomicrol (**3**) even inhibited proliferation to the extent of 84% of control. In addition, both compounds were identified as potent FLT3 inhibitors and thus display promising lead structures for further drug development. Moreover, our results confirmed the chemopreventive properties of flavonoids in general, and in particular of polymethylated flavones, which have been intensively investigated especially over the last decade.

## 1. Introduction

Multiple myeloma (MM) is a hematologic malignancy where mature B-cells, i.e., plasma cells, proliferate extensively within the bone marrow [1]. Myeloma cells display a variety of genetic aberrations and, especially upon treatment, resistant clones are selected which finally leads to disease relapse in most patients. Likewise, acute myeloid leukemia (AML) is a heterogenous hematologic disorder caused by multiple genetic abnormalities that occur in the myeloid precursor cells within the bone marrow [2]. Both diseases predominantly affect the bone marrow of patients. Although several treatment lines are available for myeloma and AML, relapses are usual and most patients finally succumb to their disease. Thus, novel therapies are urgently needed.

In our ongoing search for new lead compounds in the treatment of hematologic cancers [3,4,5], we investigated the aerial parts of *Dracocephalum moldavica* L. (Lamiaceae), one of 74 accepted species of the dragonhead genus [6]. *D. moldavica* can be found on mountains and in semiarid areas of Europe and Asia, where it has been used as a folk medicine for treating various chronic diseases, such as cardiac disease, hypertension, atherosclerosis, asthma, and other oxidative-stress related disorders [7,8,9,10]. Extensive phytochemical studies on its metabolome revealed a plethora of chemical constituents from different compound classes, including alkaloids, coumarins, cyanogenic glucosides, flavonoids, phenylpropanoids, polysaccharides, and terpenoids, which showed anti-inflammatory, antioxidant, and antitumor activities [10,11,12,13,14,15]. The variety of potential drug candidates and the activity of a crude acetone extract against Molm-13 and KMS-12-PE cells prompted us to further investigate this interesting plant species. In the present work, 13 constituents from four different compound classes were isolated and subsequently evaluated for their effects on the above-mentioned myeloma and AML cell lines.

## 2. Results and Discussion

### 2.1. Isolation and Identification

Exhaustive extraction followed by liquid–liquid partitioning and repeated chromatographic separation led to the isolation of six flavones (**1**–**6**), two flavonols (**7** and **8**), two lignans (**9** and **10**), and three caffeic acid derivatives (**11**–**13**) (Figure 1, Appendix A). After comparison of MS and NMR data and optical rotation values (see Supporting Information) with the values reported in the literature, the compounds were identified as cirsimaritin (**1**) [16], 5-desmethylsinensetin (**2**) [17], xanthomicrol (**3**) [18], apigenin (**4**) [19], luteolin (**5**) [20], diosmetin (**6**) [21], kaempferol (**7**) [19], quercetin (**8**) [22], (+)-piperitol (**9**) [23], (+)-9α-hydroxysesamin (**10**) [24], caffeic acid (**11**) [25], (R)-(+)-rosmarinic acid (**12**) [26], and (R)-(+)-3′-O-methylrosmarinic acid (**13**) [27].

### 2.2. Biological Activity

All isolated substances were tested at different concentrations for their potential to induce programmed cell death (apoptosis) in myeloma cells and AML cells (Figure 2, Table 1).

While no cytotoxic effect was found for lignans and caffeic acid derivatives, five out of six flavones (**1** and **3**–**6**) and both flavonols (**7** and **8**) induced apoptosis in both cancer cell types in a dose-dependent and time-dependent manner. In the KMS-12-PE cell line, the effects were more pronounced for the flavone compounds (except **2**) generally being more apoptotic than the flavonols, which were not active at the lowest concentration. With an EC_50_ value of 26 µM, diosmetin (**6**) was the most potent compound on the KMS-12-PE cell line.

For the AML cell line Molm-13, the overall picture was similar, also showing stronger apoptotic effects for the flavone components. However, even more evident was the higher activity of the two polymethylated derivatives, cirsimaritin (**1**) and xanthomicrol (**3**), compared to the other flavone (and flavonol) derivatives, with EC_50_ values of around 22 µM (**1**) and 28 µM (**3**). The latter two compounds were further investigated at lower concentrations in proliferation assays (Figure 3).

At concentrations ranging from 10 to 2.5 µM, cirsimaritin (**1**) and xanthomicrol (**3**) induced an inhibition of proliferation, which was marginally in the myeloma cell line but clearly induced in AML cells. Here, cirsimaritin (**1**) was inhibiting proliferation to an extent of 88 to 72% compared to untreated cells, while xanthomicrol (**3**) showed an inhibition of up to 93% (and of still 84% at a concentration of 2.5 µM). The FLT3 inhibitor gilteritinib was added as a positive control and showed high effect on AML cells but not on myeloma cells. These clear anticancer effects on AML cells at reasonably low and certainly further titratable concentrations are promising and warrant further testing.

Polymethylated flavones were previously found to inhibit the FLT3 pathway in a study on 45 natural and synthetic flavonoids using AML cell lines Molm-13 and MV-4-11 [27]. This specific potential to inhibit FLT3 is of major interest for AML. Cells harboring an activating mutation in FLT3 represent approximately one third of the AML cases, and activation of FLT3 is a risk factor for high relapse and bad prognosis [28,29]. Gilteritinib, a second-generation FLT3 inhibitor, is approved for relapsed/refractory FLT3 + AML. However, the FLT3 pathway can also play an important role in multiple myeloma.

In our previous work [30], we found that MM patients with advanced disease showed high levels of FLT3 ligand in the blood and bone marrow. In addition, we could show that the FLT3-receptor is overexpressed in a subgroup of MM patients and this overexpression correlated with inferior prognosis [31]. Inhibitors of FLT3 (midostaurin, gilteritinib) demonstrated anti-myeloma activity in vitro. Therefore, we are especially interested in novel inhibitors of FLT3 and further testing of xanthomicrol (**3**) and cirsimaritin (**1**) on primary myeloma cells is guaranteed. The lack of significant effects of compound **1** and the smaller effect of compound **3** on the tested MM cell line (KMS-12-PE) compared to AML cells might be explained by its lower dependence on the FLT3 pathway. This is corroborated by the fact that also second generation FLT3 inhibitor, gilteritinib, had a lower effect on this myeloma cell proliferation.

Regarding the structural requirements for FLT3 inhibition, Yen et al. investigated five subclasses of flavonoids [27]. Thereby, the average inhibitions were highest for flavonols, flavones, and chalcones, all of which show a planar chromone substructure. In contrast, the tested flavanones and prenylflavanones were not effective. Apart from the planar structure, several other necessary structural features for FLT3 inhibition were identified by molecular docking studies, such as a carbonyl group in position 4, a hydroxy group in position 3 or 5, a hydroxy or methoxy group in position 4′, as well as a hydroxy group in position 7. Except for the last point, all of these structural features are fulfilled by cirsimaritin (**1**) and xanthomicrol (**3**). Therefore, we investigated both compounds on their potential to reduce FLT3 kinase activity. Cirsimaritin (**1**) and xanthomicrol (**3**) effectively inhibited FLT3 kinase at concentrations of 25 to 6.25 nM, with inhibition rates of 81% (**1**) and 88% (**3**) of 20 ng protein at the lowest concentration.

With the results of our study, we could identify two novel FLT3 inhibitors of flavonoid origin. Moreover, additional findings can now be added to the structural requirements for FLT3 binding. One such finding becomes evident while looking at the structure of compound **2**, which differs from salvigenin, the second most active compound in the study of Yen et al. [27], by only one methoxy group in position 3 and which showed no effects in any of the tested cell lines. The negative effect of this additional methoxy group is corroborated by the results of the other flavonoids with this feature in the study of Yen et al., which also showed much lower activities.

An even more interesting structural feature is indicated by xanthomicrol (**3**), which possesses an additional methoxy group in position 8. This specific feature did not seem to reduce the cytotoxic effects and therefore the structure of xanthomicrol (**3**) might as well be proposed as a valuable lead compound for further optimization. Not only because this specific structural feature was missing in the study by Yen et al., but because methoxylated flavonoids are not as common as their unmethoxylated counterparts, and those bearing several methoxy groups are even more rare [32].

This particular subgroup of flavonoids, which is also referred to as polymethylated or polymethoxylated flavonoids (PMFs), became of increased interest over the last decade [32,33]; firstly, because many PMFs exhibit pronounced cancer chemopreventive properties and secondly, because they show a dramatically increased bioavailability compared to unmethylated flavonoids [34,35]. The higher bioavailability of PMFs results from a lower polarity and thus an elevated membrane penetration, as well as an increased metabolic stability caused by hindered glucuronidation and sulfation processes [35].

Nevertheless, unmethylated or monomethylated flavonoids, such as diosmetin (**6**), also contribute towards cancer chemoprevention. Even though they may show lower effects in vitro and decreased bioavailability, they are much more abundant in the plant kingdom and in our daily nutrition and are therefore consumed in significantly higher amounts. Thus, not only the discovery of a new potent lead structure for AML treatment (**3**), but also the results obtained for the more common flavonoids (**4** to **8**) are of interest. Following our report on the activity of apigenin (**4**) and luteolin (**5**) and some of their glycosides against myeloma cell lines NCI-H929, U266, and OPM2 [5], our present study demonstrates their activity against KMS-12-PE cells and the AML cell line Molm-13. In addition, cytotoxic effects of the highly abundant flavonols kaempferol (**7**) and quercetin (**8**) are presented in our study.

Due to their high abundance and availability, these compounds have been the target of repeated investigations for their cytotoxic and anticancer effects [36]. Apart from their antioxidant properties, apigenin (**4**) and quercetin (**8**) were found to modulate a number of signaling pathways involved in carcinogenesis, with apigenin also being suggested as a general cancer medication [37,38,39]. In addition, for luteolin (**5**), interesting antitumor effects have been discovered, which showed the compound to suppress metastasis in breast and colorectal cancer cells [40,41].

Summarizing, our study reveals new data for the chemopreventive effects of several prominent and some more particular flavonoids against multiple myeloma and acute myeloid leukemia. While diosmetin (**6**) was effective against myeloma cell line KMS-12-PE, two compounds (**1** and **3**) showed pronounced effects on AML cell line Molm-13, also at lower concentrations. Thereby, the latter two compounds, namely cirsimaritin (**1**) and xanthomicrol (**3**), were identified as novel FLT3 inhibitors.

## 3. Materials and Methods

### 3.1. Plant Material, Reagents and Experimental Procedures

Dried aerial parts (leaves and flowers) of *D. moldavica* were obtained from Dr. Vasilica Onofrei of the University of Agricultural Sciences and Veterinary Medicine, Faculty of Agriculture, in Iaşi, Romania. LC–MS grade acetonitrile and water and other (analytical grade) solvents and reagents were purchased from VWR International GmbH (Darmstadt, Germany). LC–MS grade formic acid was obtained from Sigma Aldrich Co. (St. Louis, MO, USA). Water used for isolation was twice distilled in-house. DMSO-*d*_6_ (99.80%, Lot S1051, Batch 0119E) and MeOH-*d*_4_ (99.80%, Lot P3021, Batch 1016B) for NMR spectroscopy were purchased from Euriso-top GmbH, Saarbrücken, Germany. TLC was performed on silica gel 60 F254 plates (VWR International, Darmstadt, Germany) using toluene-ethyl acetate-formic acid (5.5:3.5:1) as the mobile phase and vanillin-sulphuric acid for detection. Flash chromatography was carried out with a Büchi PrepChrom C-700 chromatograph using a FlashPure EcoFlex Silica Gel SL cartridge (100 g/135 mL, irregular 40–63 µm particle size, Büchi Labortechnik GmbH, Essen, Germany). Column chromatography was performed with Sephadex LH-20 (GE Healthcare AB, Uppsala, Sweden). Semi-preparative HPLC was carried out on a Waters Alliance e2695 Separations Module coupled to a 2998 Photodiode Array detector and a WFC III fraction collector using a Phenomenex Aqua column (5 µm. 250 × 10.0 mm). Extracts, fractions, and pure compounds were analyzed on a Shimadzu Nexera 2 liquid chromatograph connected to an LC–MS triple quadrupole mass spectrometer with electrospray ionization (Shimadzu, Kyoto, Japan). A Phenomenex Luna Omega C18 column (100 × 2.1 mm, 1.6 µm particle size, Phenomenex, Aschaffenburg, Germany) was used for separation. 1D (^1^H, ^13^C) and 2D (HSQC, HMBC, COSY) NMR spectra were recorded on a Bruker Avance III 400 NMR spectrometer operating at 400 MHz for the proton channel and 100 MHz for the ^13^C channel with a 5 mm PABBO broad band probe with a z gradient unit at 298 K (Bruker BioSpin GmbH, Rheinstetten, Germany). Reference values were 2.50 (^1^H) and 39.51 ppm (^13^C) for dimethyl sulfoxide as well as 3.31 (^1^H) and 49.15 ppm (^13^C) for MeOH, respectively. Structure elucidation and spectra simulations were performed using the Topspin 3.6 software (Bruker Biospin GmbH, Rheinstetten, Germany). 5 mm NMR sample tubes were obtained from Rototec-Spintec GmbH, Griesheim, Germany. The specific rotation of compounds was measured on a Jasco P-2000 polarimeter (Jasco, Pfungstadt, Germany).

### 3.2. Extraction and Isolation

1140 g of dried plant material were ground and extracted five times with 6 L of an 85% aqueous acetone solution using ultrasonication followed by 24 h of maceration. The acetone was evaporated under reduced pressure and the remaining aqueous solution (2224 mL) was extracted five times with 500 mL of dichloromethane to afford 9.021 g of extract. The extract was subjected to flash chromatography using silica gel as the stationary phase and a mixture of *n*-hexane (A) and acetone (B) as the mobile phase with the following gradient: 1%B to 2%B in 10 min, to 5%B in 10 min, to 10%B in 10 min, to 20%B in 10 min, to 33%B in 10 min and to 50%B in 40 min. Of the resulting ten fractions (A–J), fractions H (1389 mg) and J (1097 mg) were further processed. Fraction H was subjected to Sephadex LH-20 chromatography (100 × 3 cm) and methanol as the solvent to give eleven fractions (H1–H11). Fraction H9 (30.74 mg) was subjected to semi-preparative chromatography using a mixture of 0.025% formic acid and acetonitrile (55:45) to give 3.30 mg of 5-desmethylsinensetin (**2**), 11.99 mg of xanthomicrol (**3**), and 3.69 mg of (+)-piperitol (**9**). Fraction H10 (16.58 mg) was chromatographed in the same manner to yield 3.46 mg of cirsimaritin (**1**), 8.6 mg of 9α-hydroxysesamin (**10**), and another 2.93 mg of xanthomicrol (**3**). Fraction J was also subjected to Sephadex LH-20 chromatography (100 × 3 cm) and methanol as the solvent to give 16 fractions (J1–J16). Fraction J12 (30.80 mg) was subjected to semi-preparative chromatography using a mixture of 0.025% formic acid and acetonitrile (70:30) to give 4.06 mg of caffeic acid (**11**), 3.31 mg of rosmarinic acid (**12**), and 5.44 mg of 3-methylrosmarinic acid (**13**). Fraction J14 (25.19 mg) was subjected to semi-preparative chromatography using a mixture of 0.025% formic acid and acetonitrile (65:35) to yield 3.02 mg of apigenin (**4**), 2.86 mg of luteolin (**5**), and 4.85 mg of diosmetin (**6**). Fraction J15 (12.74 mg) was subjected to semi-preparative chromatography using a mixture of 0.025% formic acid and acetonitrile (55:45) to give 3.25 mg of kaempferol (**7**) and another 4.38 mg of luteolin (**5**). Fraction J16 (16.29 mg) was subjected to Sephadex LH-20 chromatography (100 × 1 cm) using methanol as the solvent yielding 11.47 mg of quercetin (**8**).

### 3.3. Cytotoxicity Assays, Proliferation Assays, and FLT3 Kinase Assay

Cytotoxicity was measured as induction of apoptosis in myeloma cell line KMS-12-PE and AML cell line Molm-13, staining the cells with AnnexinV-fluorescein isothiocyanate (AnnexinV-FITC) and propidium iodide (PI). Cell lines were purchased from DSMZ (Braunschweig, Germany) and routinely fingerprinted and tested for mycoplasma negativity. All cell lines were grown in RPMI-1640 medium (Life Technologies, Paisley, UK), and supplemented with 10% fetal calf serum (FCS; PAA, Linz, Austria), L-glutamine 100 µg/mL, and penicillin-streptomycin 100 U/mL. Compounds were dissolved in DMSO at a stock concentration of 50 mM and stored at −20 °C. Briefly, 0.5 × 10^6^ cancer cells/mL were incubated for 24 h and 48 h with or without the tested compounds at indicated concentrations. Analyses were performed in duplicates and a solvent control was included. The extent of non-apoptotic cells (AnnexinV/propidium iodide negativity) was calculated as the percentage of viable cells in respect to the untreated control. Data are shown as the mean percentage of viable cells +/− standard deviation (SD) (error bars).

Proliferation was measured using a modified MTT assay (EZ4U kit, Biomedica, Vienna, Austria) according to the manufacturer’s instructions. In brief, 2.5 × 10^4^ (Molm-13)—5.0 × 10^4^ cells (KMS-12-PE cells) were seeded in 96-well plates and substances were added as indicated. Different concentrations are due to the different doubling time of the cells (24 h vs. 48 h). Cells were incubated for 48 h and during the last 7 h of incubation, 3-(4,5-dimethylthiazol-2-yl)-2,5-diphenyltetrazolium bromide was added as a substrate. Reduction of the tetrazolium salt to formazan by the mitochondrial activity of the growing cells was measured as optical density at 492 nm (with 620 nm as reference) on a plate reader. Proliferation (mean +/− SD) was calculated as the percentage of control (without substances). Concentrations used were 10 µM/5 µM/2.5 µM for all substances. Gilteritinib was used as a positive control at 2.5 µM.

FLT3 inhibitory activity of compounds **1** and **3** was determined using the Z’-LYTE screening protocol. Z’-LYTE Kinase Assay—Tyrosine 2 Peptide Kit and FLT3 were purchased from Thermo Fisher Scientific Inc. (Waltham, MA, USA). The assay was performed according to the manufacturer’s instructions using a kinase buffer (50 mM HEPES pH 7.5, 0.01% BRIJ-35, 10 mM MgCl_2_, 1 mM EGTA) containing 0.01 to 20 ng of FLT3. Compounds were measured at concentrations of 50 nM/25 nM/12.5 nM/6.25 nM using gilteritinib as a positive control.

## 4. Conclusions

In the present study, a series of phenolic compounds was studied for their activity against myeloma cell line KMS-12-PE and AML cell line Molm-13. Of the 13 tested compounds, five out of six flavones (**1**, **3**–**6**), and the two investigated flavonols, kaempferol (**7**) and quercetin (**8**), induced apoptosis in a dose-dependent manner, confirming once more the broad chemoprotective potential of flavonoids in carcinogenesis. The two polymethylated flavones, cirsimaritin (**1**) and xanthomicrol (**3**), were further examined for their antiproliferative effects at lower concentrations and showed clear impact on the AML cell line. Subsequent experiments revealed both compounds to effectively inhibit FLT3 kinase activity, which will be further examined in future studies.

## Figures and Tables

**Figure 1 ijms-23-14219-f001:**
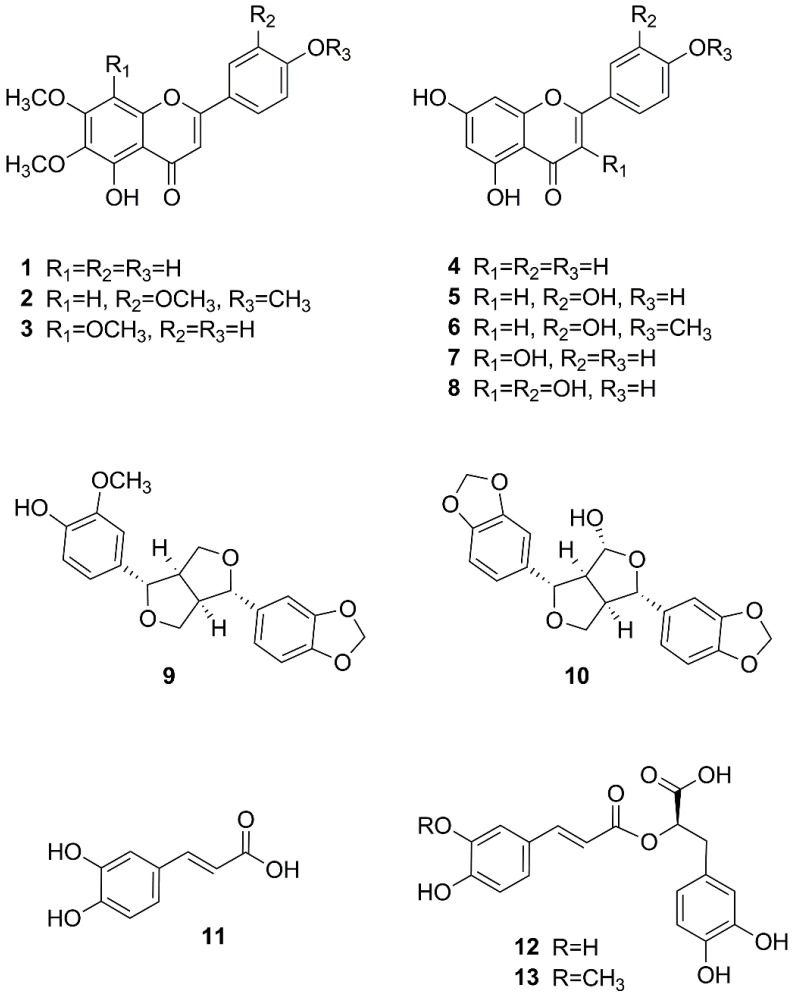
Chemical structures of compounds isolated from *D. moldavica*.

**Figure 2 ijms-23-14219-f002:**
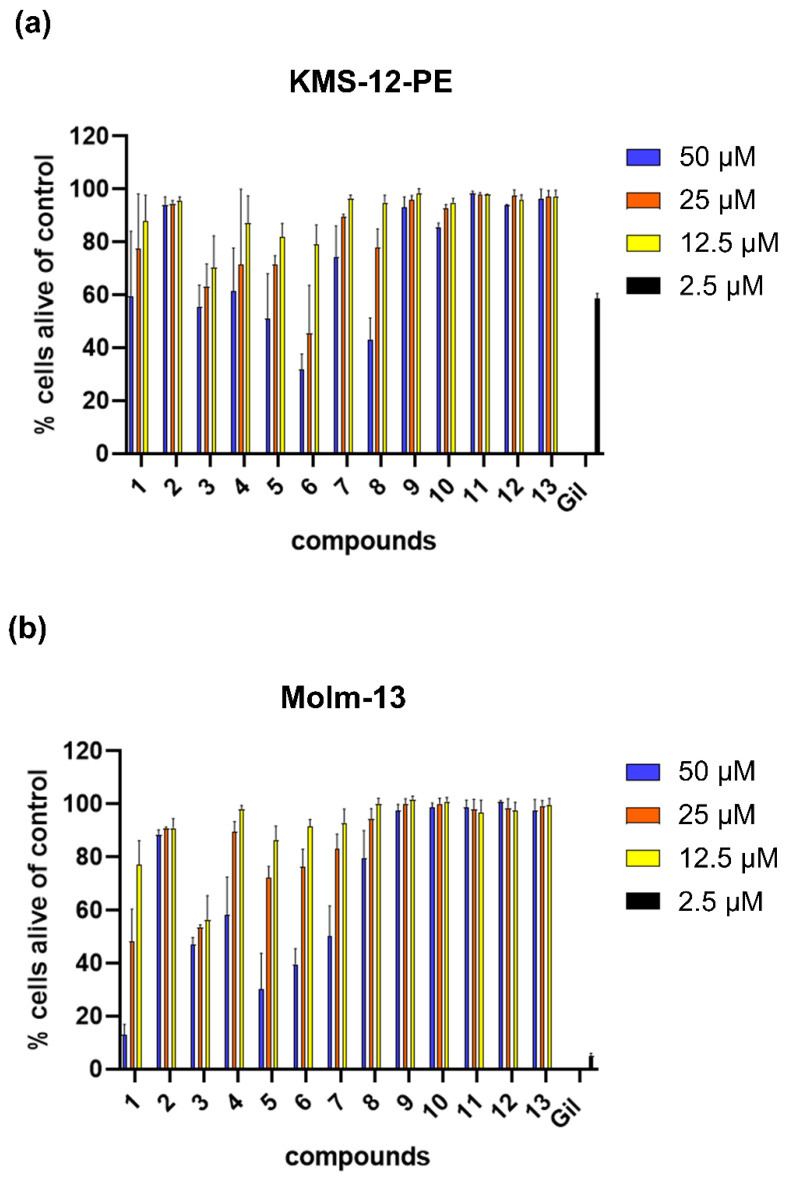
Apoptosis of cancer cells after treatment with isolated compounds. Percentage of cells alive after treatment with compounds **1** to **13**: (**a**) KMS-12-PE and (**b**) Molm-13 cells were treated with indicated concentrations (50 µM/25 µM/12.5 µM) of compounds for 48 h. Gilteritinib (2.5 µM) was used as positive control. Cell survival was measured by calculating cells which did not stain with AnnexinV/PI in comparison to untreated controls. Mean +/− SD of 2–4 independent experiments in duplicates are shown.

**Figure 3 ijms-23-14219-f003:**
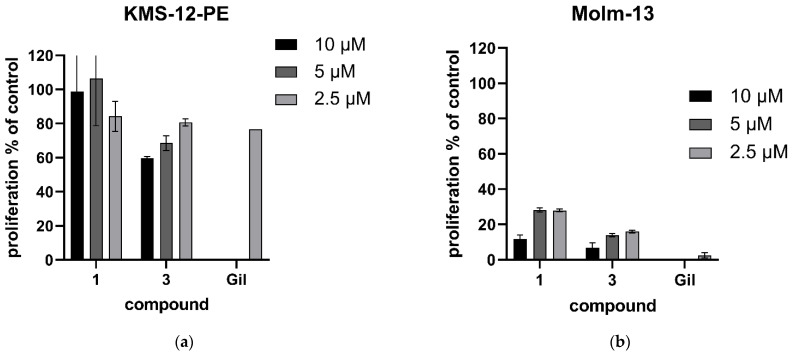
Proliferation of cancer cells after treatment with selected compounds. Percentage of proliferation after treatment with different compounds for 48 h was measured with a modified MTT assay. (**a**) KMS-12-PE and (**b**) Molm-13 cells were treated with indicated concentrations (10 µM/5 µM/2.5 µM) of compounds **1** and **3** and gilteritinib (2.5 µM) as positive control. Percentage of proliferation was calculated compared to untreated (set at 100%). Mean +/− SD of 3 (AML)—2 (MM) independent experiments are shown.

**Table 1 ijms-23-14219-t001:** Results of apoptosis measurements on KMS-12-PE and Molm-13 cells after treatment with different compounds and concentrations for 48 h. Results are given in percentage of cells alive of control. Number of independent experiments is given in parentheses. EC_50_ values were calculated using the “best-fit” model and are given in µM.

		KMS-12-PE		Molm-13	
Compound	Concentration	% Alive	EC_50_	% Alive	EC_50_
**1**	50 µM	58.9 ± 28.6 (4)		12.9 ± 4.1 (4)	
	25 µM	76.6 ± 25.6 (4)	>50	48.4 ± 12.0 (4)	21.74
	12.5 µM	86.8 ± 14.5 (4)		77.3 ± 9.0 (4)	
**2**	50 µM	93.1 ± 10.2 (3)		88.6 ± 1.8 (2)	
	25 µM	93.4 ± 9.3 (3)	>50	90.9 ± 0.4 (2)	>50
	12.5 µM	94.6 ± 10.2 (3)		91.1 ± 3.4 (2)	
**3**	50 µM	55.4 ± 13.1 (4)		47.0 ± 2.7 (4)	
	25 µM	62.6 ± 13.1 (4)	45.39	53.5 ± 1.0 (4)	27.98
	12.5 µM	69.5 ± 16.0 (4)		56.4 ± 9.1 (4)	
**4**	50 µM	59.6 ± 16.1 (4)		58.4 ± 14.2 (4)	
	25 µM	69.4 ± 31.3 (4)	>50	89.5 ± 3.9 (4)	>50
	12.5 µM	85.5 ± 10.6 (4)		98.0 ± 1.6 (4)	
**5**	50 µM	50.3 ± 18.4 (4)		30.2 ± 13.7 (4)	
	25 µM	70.5 ± 5.6 (4)	>50	72.4 ± 4.0 (4)	42.31
	12.5 µM	80.3 ± 5.4 (4)		86.5 ± 5.2 (4)	
**6**	50 µM	31.1 ± 5.3 (4)		39.8 ± 5.8 (4)	
	25 µM	44.2 ± 19.0 (4)	25.65	76.2 ± 6.7 (4)	>50
	12.5 µM	77.3 ± 6.9 (4)		91.5 ± 2.8 (4)	
**7**	50 µM	76.3 ± 17.3 (3)		50.2 ± 11.4 (4)	
	25 µM	91.2 ± 15.0 (3)	>50	83.1 ± 5.7 (4)	>50
	12.5 µM	98.0 ± 2.7 (3)		93.1 ± 5.1 (4)	
8	50 µM	43.7 ± 8.3 (3)		79.8 ± 10.2 (4)	
	25 µM	79.0 ± 4.8 (3)	>50	94.4 ± 3.9 (4)	>50
	12.5 µM	96.6 ± 2.7 (3)		100.1 ± 2.2 (4)	
**9**	50 µM	97.1 ± 4.1 (2)		97.5 ± 2.4 (2)	
	25 µM	100.1 ± 0.8 (2)	>50	100.0 ± 2.1 (2)	>50
	12.5 µM	102.8 ± 0.8 (2)		101.8 ± 1.3 (2)	
**10**	50 µM	89.4 ± 0.8 (2)		98.7 ± 1.7 (2)	
	25 µM	96.7 ± 0.5 (2)	>50	100.2 ± 2.0 (2)	>50
	12.5 µM	99.0 ± 0.6 (2)		101.1 ± 1.4 (2)	
**11**	50 µM	102.6 ± 0.5 (2)		98.8 ± 2.8 (2)	
	25 µM	102.5 ± 0.9 (2)	>50	98.1 ± 3.8 (2)	>50
	12.5 µM	102.3 ± 1.8 (2)		96.9 ± 4.6 (2)	
**12**	50 µM	98.2 ± 1.5 (2)		100.8 ± 0.6 (2)	
	25 µM	101.6 ± 1.5 (2)	>50	98.5 ± 3.6 (2)	>50
	12.5 µM	100.2 ± 0.9 (2)		97.6 ± 3.2 (2)	
**13**	50 µM	100.7 ± 3.4 (2)		97.8 ± 4.0 (2)	
	25 µM	101.3 ± 1.8 (2)	>50	99.4 ± 1.9 (2)	>50
	12.5 µM	101.5 ± 1.6 (2)		99.6 ± 2.6 (2)	

## Data Availability

Not applicable.

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
