# Peer review of "Flavones, Flavonols, Lignans, and Caffeic Acid Derivatives from Dracocephalum moldavica and Their In Vitro Effects on Multiple Myeloma and Acute Myeloid Leukemia"

_ijms, 2022, doi:10.3390/ijms232214219_

Round 1

Reviewer 1 Report

MAJOR POINT

The effect of flavones and flavonoids on the MM and AML lines is already described in recent literature. The only novelty reported in this work is represented by the effect of compound 3 on apoptosis and cell proliferation in this experimental model, too little to deserve publication on IJMS. The possible mechanism of action should be investigated. In particular, as suggested by the authors, the possible role as an inhibitor of FLT3 should be explored which, from the results presented in the present version of the work, does not yet emerge.

In order to assert by the authors in the conclusion section that “compound 3 could be a promising molecule for the development of FLT3 inhibitors” (in lines 268-270), to mention the structural similarity of 3 to compound 1 is not enough. Therefore, the authors will have to show that (a ) compound 3 has a binding site on FLT3, through molecular docking experiments, and (b) demonstrate that compound 3 is a putative inhibitor of FLT3 through an in vitro functional assay such as the Z'-LYTE kinase assay.

Minor points

(a) In table, it is not clear whether the EC50 value refers only to the Molm13 or is an average of the results obtained in the two lines. In the first case, the EC50 value of the KMS12PE must also be indicated. In the other case, the SD should be indicated.

(b) In table 1, the EC50> 50 for compound 6 does not appear to be real in the light of the results shown alongside (% of alive) and should be corrected or clarified. A fair value seems to be more or less 30 microM, a value very close to that of compound 1 and 3. If so, why haven't you further processed compound 6 as well by testing its antiproliferative activity?

(c) To describe apoptosis the authors stated in line 242 that "the extent of non-apoptotic cells (AnnexinV/7AAD negativity) was calculated as percentage of viable cells in respect to the untreated control". In other words, they reported in Fig.2 the percentage of cells that do not stain in the detection procedure (alive, AnnV negative and 7-AAD negative) rather than those stained by the reagent. To dispel the doubt of an experimental artifact, such as that cells evaluated as live could be only not stained by the reagent, it is mandatory to insert a positive control in Fig. 2, showing i.e. the apoptotic effect of gilteritinib and midostaurin.

(d) Line 98… cytostatic effects on AML cells? Is it right?

(e) Cytotoxicity assays and proliferation assay section, page 7. The solvent used to suspend the compounds, the concentration of the stock solution, the preservation and storage methods of the stock solutions were never indicated.

(f) The authors should explain why compounds 1 and 3 have comparable effects on apoptosis (NB, the EC50s are similar in Molm13 and KMS12PE) but have distinct antiproliferative effects on the two cell lines.

Author Response

MAJOR POINT

The effect of flavones and flavonoids on the MM and AML lines is already described in recent literature. The only novelty reported in this work is represented by the effect of compound 3 on apoptosis and cell proliferation in this experimental model, too little to deserve publication on IJMS. The possible mechanism of action should be investigated. In particular, as suggested by the authors, the possible role as an inhibitor of FLT3 should be explored which, from the results presented in the present version of the work, does not yet emerge.

In order to assert by the authors in the conclusion section that “compound 3 could be a promising molecule for the development of FLT3 inhibitors” (in lines 268-270), to mention the structural similarity of 3 to compound 1 is not enough. Therefore, the authors will have to show that (a ) compound 3 has a binding site on FLT3, through molecular docking experiments, and (b) demonstrate that compound 3 is a putative inhibitor of FLT3 through an in vitro functional assay such as the Z'-LYTE kinase assay.

Dear Reviewer,

Thank you very much for your comments on our work. Please find our answers below.

Major point (a) – docking studies:

Yen et al. (our Ref. 27) did docking studies on the FLT3 receptor using different active and inactive flavonoid subclasses as well as the most active molecules from their in vitro screening experiments. Thereby, all critical/necessary interactions were defined as well as those that might decrease the activity but not hinder binding in general.

Compounds 1 and 3 of our study display all of the essential features for FLT3 binding. These are the planar chromone structure and the thus resulting π-alkyl interactions with L616, V624, A642, and L818 as well as π-π T-shaped interactions of the B-ring with gatekeeper F691 and DFG-loop F830. Binding to the DFG-loop is, furthermore, guaranteed by the 4’-hydroxy group, which forms an essential hydrogen bond to D829. At the same time, the B-ring of both compounds shows no other substituents that might counteract binding, as e.g. observed for compound 2 in our study. Other essential hydrogen bonds are formed by the carbonyl group in position 4 and the hydroxy group in position 5, which are crucial for binding to the hinge residue C694 and assumed as the most critical bindings by Yen et al. The only feature that is not fulfilled is the hydroxy group in position 7 that is necessary for the hydrogen bond with L616. Here, compounds 1 and 3 show a methoxy group instead, which according to Yen et al. leads to a decrease in activity but does not affect binding in general (as demonstrated by compound 30 of their study).

Major point (b) – FLT3 assays:

In order to adhere to the second major comment of the reviewer and to prove the FLT3 inhibition of our compounds in vitro, we ordered the Z’-LYTE kinase assay. Thereby, we could show that both compounds (as well as the positive control gilteritinib) inhibited the FLT3 kinase until a concentration of 6.25 nM. We added this finding into our manuscript. Due to the long shipping time of the assay and the approaching of the (already extended) deadline, we were not able to extensively characterize the FLT3 inhibition of both compounds. However, with the assay now being set up, we will continue our research on this interesting target in a follow-up study.

The reason, why we did not use the assay in the first place is that we were not specifically looking for novel FLT3 inhibitors, but for new lead compounds for the treatment of AML and multiple myeloma. We therefore chose a plant species, which we found active in preliminary experiments and which is known for its abundance in phenolic compounds. However, we did not know in the beginning, which particular compounds would come out at the end of the isolation procedure, a fact that is quite common in phytochemistry. 

Therefore, we conducted measurements of apoptosis and antiproliferative assays which led to the identification of compounds 1 and 3 as potent agents on the Molm-13 cell line. The inhibitory activity of FLT3 by the two substances is a subsequent result from literature studies and the reason why no kinase assays were initially performed. However, with our in vitro results we now can definitely say that the compounds do not only possess the necessary features for FLT3 binding but also exhibit potent inhibitory effects.

Minor points

(a) In table, it is not clear whether the EC50 value refers only to the Molm13 or is an average of the results obtained in the two lines. In the first case, the EC50 value of the KMS12PE must also be indicated. In the other case, the SD should be indicated.

A: We are sorry for the confusion. EC50 values were calculated for Molm-13 activity. We did not calculate EC50 values for KMS-12-PE because of the lower number of measurements and the thus low validity of the results. However, while revising our manuscript, we conducted additional experiments on KMS-12-PE and obtained enough data to also calculate valid EC50 values (see revised table).

(b) In table 1, the EC50> 50 for compound 6 does not appear to be real in the light of the results shown alongside (% of alive) and should be corrected or clarified. A fair value seems to be more or less 30 microM, a value very close to that of compound 1 and 3. If so, why haven't you further processed compound 6 as well by testing its antiproliferative activity?

A: The reviewer is right. At the first look, it seems that the EC50 value of compound 6 has to be between 25 and 50 µM. However, we used the model of „Best-fit values” for calculation of the EC50s, which takes also standard deviations into account and calculates the most appropriate curve. Therefore, the obtained results do not always result in the expected concentrations, as e.g. for compound 6, which has an EC50 of 55 µM. We added this information into the manuscript.

 (c) To describe apoptosis the authors stated in line 242 that "the extent of non-apoptotic cells (AnnexinV/7AAD negativity) was calculated as percentage of viable cells in respect to the untreated control". In other words, they reported in Fig.2 the percentage of cells that do not stain in the detection procedure (alive, AnnV negative and 7-AAD negative) rather than those stained by the reagent. To dispel the doubt of an experimental artifact, such as that cells evaluated as live could be only not stained by the reagent, it is mandatory to insert a positive control in Fig. 2, showing i.e. the apoptotic effect of gilteritinib and midostaurin.

A: We inserted the effect of gilteritinib as a positive control as suggested by the reviewer.

(d) Line 98… cytostatic effects on AML cells? Is it right?

A: Thanks for pointing out this wording – since we show apoptosis until 12.5µM and the antiproliferative effects starting at 10µM we cannot clearly distinguish between cytotoxic and cytostatic effects at this point. Therefore, we changed the wording to “anticancer effects”.

(e) Cytotoxicity assays and proliferation assay section, page 7. The solvent used to suspend the compounds, the concentration of the stock solution, the preservation and storage methods of the stock solutions were never indicated.

A: All details (solvent: DMSO, stock solution 50mM, storage at -20°C) were now added to the materials and methods section.

(f) The authors should explain why compounds 1 and 3 have comparable effects on apoptosis (NB, the EC50s are similar in Molm13 and KMS12PE) but have distinct antiproliferative effects on the two cell lines.

A: Compounds 1 and 3 also have distinct effects on the apoptosis, which was, however, not clear to the previously missing EC50 values against the KMS-12-PE cell line. This shortcoming was corrected.

Reviewer 2 Report

Introduction

The authors briefly present an introduction to the research problem. They describe the plant they investigated, its use, and the families of compounds that are present in plant. Then they go to the heart of the matter, which is their research intention. I have no objections to this part.

Results and discussion

What specific compounds were isolated and the source literature for their identification were briefly presented. Present structures of compounds facilitate the reception of part 2.1. Next, the authors present the results of cell vialbity treated with different concentrations of individual compounds. Table 1 lists specific values ​​for each experiment. I would recommend expanding "cpd" and "conc" to full names. In addition, I have doubts about the markings in the Table 1. In the case of the KMS12PE line the values ​​refer to the cell survival (% alive)? So how is it also described under the name of the Molm13 line? I am asking for an explanation and possible appropriate improvement. Additionally, the presented EC50 values ​​refer to which line - KMS12PE or Molm13? Please explain and possibly improve. Moreover, what is correct: Molm-13 (line 47) or the one in Table 1, the same as KMS-12-PE (line 47) or the one in Table 1? Please standardize. The rest of this section is written correctly. A very well-done analysis of the key substituents and their place of substitution affecting the activity. A similar analysis was also carried out in DOI: 10.3390/molecules24040679 regarding the substitution of naringenin with alkyl chains (not always the compounds with OH groups were the most active), and DOI: 10.1007/s00044-017-1887-9 regarding the substitution of OH groups in xanthohumol with acyl groups (decrease in activity ). Certainly, it would be good if the authors of the reviewed article wanted to continue their research on other lines as well.

Materials and methods

This section has a standard layout. I also have some questions and suggestions. Was the humidity of the dried material tested? In line 196, please correct "13C". What on line 207 means "(85%)" for acetone? Does that mean it was an 85% aqueous acetone solution? I would also recommend creating a graph showing the isolation (fractions, solvents/eluents, what compound is isolated). It can be added to Supplementary Materials. This would make it easier to pick up and understand the entire isolation/extraction process. Throughout subsection 3.2, compound numbers should be in bold. Line 239 should be "mL". Lines 255-256 are missing spaces between numbers and units.

Conclusions

Conclusions arise directly from the research and from the discussion of the results. I have no objections.

Author Response

Introduction

The authors briefly present an introduction to the research problem. They describe the plant they investigated, its use, and the families of compounds that are present in plant. Then they go to the heart of the matter, which is their research intention. I have no objections to this part.

Results and discussion

What specific compounds were isolated and the source literature for their identification were briefly presented. Present structures of compounds facilitate the reception of part 2.1. Next, the authors present the results of cell vialbity treated with different concentrations of individual compounds. Table 1 lists specific values ​​for each experiment.

I would recommend expanding "cpd" and "conc" to full names.

A: Both abbreviations were expanded, now reading “compounds” and “concentrations”.

In addition, I have doubts about the markings in the Table 1. In the case of the KMS12PE line the values ​​refer to the cell survival (% alive)? So how is it also described under the name of the Molm13 line? I am asking for an explanation and possible appropriate improvement.

A: We apologize for the confusion. Yes, also for the KMS-12-PE cell line results are given in % alive of control. We added the missing subheading for this column.

Additionally, the presented EC50 values ​​refer to which line - KMS12PE or Molm13? Please explain and possibly improve.

A: EC50 values were calculated for the Molm-13 cell line, as here more values below 50µM were observed. We also added the EC50 values against the KMS-12-PE cell line after now having conducted enough measurements. We are sorry for the confusion.

Moreover, what is correct: Molm-13 (line 47) or the one in Table 1, the same as KMS-12-PE (line 47) or the one in Table 1? Please standardize.

A: It is Molm-13 and KMS-12-PE. Actually, we standardized before submission, but somehow overlooked the table heading. We apologize for the confusion.

The rest of this section is written correctly. A very well-done analysis of the key substituents and their place of substitution affecting the activity. A similar analysis was also carried out in DOI: 10.3390/molecules24040679 regarding the substitution of naringenin with alkyl chains (not always the compounds with OH groups were the most active), and DOI: 10.1007/s00044-017-1887-9 regarding the substitution of OH groups in xanthohumol with acyl groups (decrease in activity ). Certainly, it would be good if the authors of the reviewed article wanted to continue their research on other lines as well.

A: Thank you very much for you nice comments and the literature suggestions. Yes, we plan to extend our research to additional AML cell lines in the future.

Materials and methods

This section has a standard layout. I also have some questions and suggestions. Was the humidity of the dried material tested?

A: The humidity of the dried plant material was not tested but was not considered necessary for extraction purposes, even more so as water was part of the extracting solvent. However, we freeze dried the crude extract and all isolated substances prior to activity testing.

In line 196, please correct "13C".

A: This was corrected.

What on line 207 means "(85%)" for acetone? Does that mean it was an 85% aqueous acetone solution?

A: Yes, we changed this in the manuscript following your suggestion.

I would also recommend creating a graph showing the isolation (fractions, solvents/eluents, what compound is isolated). It can be added to Supplementary Materials. This would make it easier to pick up and understand the entire isolation/extraction process.

A: An isolation scheme was inserted into the Supporting Information, in which isolation steps, respective techniques and solvents, and the resulting fractions and compounds are depicted.

Throughout subsection 3.2, compound numbers should be in bold.

A: All compound numbers in this section were changed to bold style.

Line 239 should be "mL".

A: This was changed.

Lines 255-256 are missing spaces between numbers and units.

A: Spaces in lines 255-256 (and also in line 252) were added.

Conclusions

Conclusions arise directly from the research and from the discussion of the results. I have no objections.

Reviewer 3 Report

The article "Flavones, Flavonols, Lignans and Caffeic Acid Derivatives from Dracocephalum moldavica and their in vitro Effects on Multiple Myeloma and Acute Myeloid Leukemia " reports new findings worthy of publication in International Journal of Molecular Sciences. The manuscript described the 13 phenolic constituents was evaluated for its cytotoxic effects in myeloma (KMS-12-PE) and AML (Molm-13) cells in vitro by MTT assay. The compounds were identified in some previous paper and anti-cancer activity results showed IC50s more than 50 μM, except compound 1 and 3. Based on the limited mechanistic data including inhibitory effect on cell proliferation, the data suggest that compound 1 and 3 are FLT3 inhibitors. Even if the study is sound, the novelty is either insufficient or not documented enough. In conclusion, I do not recommend the publication of this manuscript in this format in the International Journal of Molecular Sciences.

Author Response

The article "Flavones, Flavonols, Lignans and Caffeic Acid Derivatives from Dracocephalum moldavica and their in vitro Effects on Multiple Myeloma and Acute Myeloid Leukemia " reports new findings worthy of publication in International Journal of Molecular Sciences. The manuscript described the 13 phenolic constituents was evaluated for its cytotoxic effects in myeloma (KMS-12-PE) and AML (Molm-13) cells in vitro by MTT assay. The compounds were identified in some previous paper and anti-cancer activity results showed IC50s more than 50 μM, except compound 1 and 3. Based on the limited mechanistic data including inhibitory effect on cell proliferation, the data suggest that compound 1 and 3 are FLT3 inhibitors. Even if the study is sound, the novelty is either insufficient or not documented enough. In conclusion, I do not recommend the publication of this manuscript in this format in the International Journal of Molecular Sciences.

Dear reviewer,

Thank you for you clear comments. We are sorry that the first version of our manuscript did not meet your expectations. However, in the last two month we did a series of additional experiments, such as repeated measurements on the KMS-12-PE cell line (which found compound 6 to be effective against MM cells) as well as in vitro studies on the FLT3 kinase (with which we could confirm the primarily putative FLT3 inhibition). We also re-investigated all of our isolated substances and found to actually have identified two novel FLT3 inhibitors. Moreover, we created a supporting information providing NMR data of all isolated components as well as optical rotation values, thus also indicating the respective enantiomer.

We hope that with the revisions made, the elucidation of the mechanism and the increased documentation of our results, you will now find our manuscript suitable for publication.

Reviewer 4 Report

This manuscript describes the isolation of various phenolic compounds from the aerial part of the Moldavian dragonhead plant used as folk medicine. 13 compounds were identified to known flavones, flavonols and lignans. The compounds were evaluated on two cell lines related to Multiple Myeloma (KMS-12-PE) and Acute Myeloma Leukemia (Molm-13). Two flavones: salvigenin and xanthomicrol displayed moderate activities on these cells.  Since salvigenin has been previously reported to be FLT3 inhibitors (Yen et al, .Ref 27) both compounds were proposed has hit compounds for further investigation as acute myeloma leukemia potential drugs.

Such polyphenol compounds are frequent hitter compounds with >10 micromolar activities on a large variety of targets and suffering of low bioavailability. In the present case only two compounds presented a moderate activity on Acute Myeloma Leukemia cell line Molm-13. The main finding of the work is that xanthomicrol is slightly more active than salvigenin once reported to be a FLT3 inhibitors (Ref 27).  This finding is worthy to be published provide the authors address the following remarks.

The main interest is that xanthomicrol might be a tyrosine kinase inhibitor of FLT-3.  However it is not demonstrated, the authors arguing structure similarity with other flavones, that is questionable point. Simple FLT3 kinase assays are commercially available, thus it is easy to the authors to settle this point.  

The identification of the structures is poorly discussed. Since there is no NMR data in Suppl. Mat.. it may be expected a suitable reference for these data. For xanthomicrol the reference provided does not include NMR data. The correct one is : Apaza T, et al. https://doi.org/10.1016/j.jep.2019.112036r

Fig 3. Representation on the same figure of the activities of the references nanomolar compounds, gilteritinib and midostaurin with the present flavones active at about ten micromolar is misleading. We just need the activity at 2.5 micromolar of gilteritinib and midostaurin to compare.

Minor remark

The black and white graphics in Fig 2 are almost unreadable. Please enlarge them presenting the diagrams one above the other or switch for a color Figure.

Author Response

This manuscript describes the isolation of various phenolic compounds from the aerial part of the Moldavian dragonhead plant used as folk medicine. 13 compounds were identified to known flavones, flavonols and lignans. The compounds were evaluated on two cell lines related to Multiple Myeloma (KMS-12-PE) and Acute Myeloma Leukemia (Molm-13). Two flavones: salvigenin and xanthomicrol displayed moderate activities on these cells.  Since salvigenin has been previously reported to be FLT3 inhibitors (Yen et al, .Ref 27) both compounds were proposed has hit compounds for further investigation as acute myeloma leukemia potential drugs.

Such polyphenol compounds are frequent hitter compounds with >10 micromolar activities on a large variety of targets and suffering of low bioavailability. In the present case only two compounds presented a moderate activity on Acute Myeloma Leukemia cell line Molm-13. The main finding of the work is that xanthomicrol is slightly more active than salvigenin once reported to be a FLT3 inhibitors (Ref 27).  This finding is worthy to be published provide the authors address the following remarks.

The main interest is that xanthomicrol might be a tyrosine kinase inhibitor of FLT-3.  However it is not demonstrated, the authors arguing structure similarity with other flavones, that is questionable point. Simple FLT3 kinase assays are commercially available, thus it is easy to the authors to settle this point. 

A: The reviewer is right, that we did not explore the possible role of xanthomicrol (3) as an FLT3 inhibitor by kinase assays, which is due to the design of our study. In contrast to e.g. Yen et al., who were looking for novel FLT3 inhibitors, our study focussed on the detection of new lead compounds for the treatment of AML and multiple myeloma. We therefore chose a plant species, which we found active in preliminary experiments and which is known for its abundance in phenolic compounds. However, we did not know in the beginning, which particular compounds would come out at the end of the isolation procedure. Therefore, we conducted measurements of apoptosis and antiproliferative assays which led to the identification of two compounds as potent agents on the Molm-13 cell line. The inhibitory activity (1) or putative inhibition (3) of FLT3 by the two substances is a subsequent result from literature studies and the reason why no kinase assays were performed. However, as written in our manuscript, further studies on xanthomicrol (3) will include mechanistic studies on FLT3 kinase.

With regard to binding on the FLT3 receptor, Yen et al. (our Ref. 27) did docking studies on the FLT3 receptor using different active and inactive flavonoid subclasses as well as the most active molecules from their in vitro screening experiments. Thereby, all critical/necessary interactions were defined as well as those that might decrease the activity but not hinder binding in general.

Xanthomicrol, compound 3 of our study, displays all of the essential features for FLT3 binding. These are the planar chromone structure and the thus resulting π-alkyl interactions with L616, V624, A642, and L818 as well as π-π T-shaped interactions of the B-ring with gatekeeper F691 and DFG-loop F830. Binding to the DFG-loop is, furthermore, guaranteed by the 4’-hydroxy group, which forms an essential hydrogen bond to D829. At the same time, the B-ring of xanthomicrol (3) shows no other substituents that might counteract binding, as e.g. observed for compound 2 in our study. Other essential hydrogen bonds are formed by the carbonyl group in position 4 and the hydroxy group in position 5, which are crucial for binding to the hinge residue C694 and assumed the most critical bindings by Yen et al. The only feature, which is not fulfilled is the hydroxy group in position 7 that is necessary for the hydrogen bond with L616. Here, xanthomicrol (3) shows a methoxy group instead, which according to Yen et al. leads to a decrease in activity but does not affect binding in general (as demonstrated by compound 30 of their study). Therefore, binding of xanthomicrol (3) can be assumed, but, as written above, will be part of our future studies.

The identification of the structures is poorly discussed. Since there is no NMR data in Suppl. Mat.. it may be expected a suitable reference for these data. For xanthomicrol the reference provided does not include NMR data. The correct one is: Apaza T, et al. https://doi.org/10.1016/j.jep.2019.112036r

A: The reviewer is right that the identification of structures was not attributed enough attention. Though in the reference we used for xanthomicrol the NMR data can be found in the supporting information, we found other references which were not adequate and exchanged them with more appropriate ones. We also added the NMR data of our compounds to the supporting information as well as optical rotation values in order to also give information on the respective enantiomer. In the course of the structural revisions, we discovered misleading results from the Reaxys database, which reports the name salvigenin for the structure of cirsimaritin and thus led us to the wrong chemical structure. We corrected both the name and the structure in the manuscript. Thanks to your comment and the subsequent revisions, our study now resulted in the identification of two FLT3 inhibitors, namely cirsimaritin and xanthomicrol.

Fig 3. Representation on the same figure of the activities of the references nanomolar compounds, gilteritinib and midostaurin with the present flavones active at about ten micromolar is misleading. We just need the activity at 2.5 micromolar of gilteritinib and midostaurin to compare.

A: We changed the representation of the positive control and now show gilteritinib at a concentration of 2.5 µM in Figures 2 and 3.

Minor remark

The black and white graphics in Fig 2 are almost unreadable. Please enlarge them presenting the diagrams one above the other or switch for a color Figure.

A: Figs. 2a and 2b were enlarged and presented above each other. Additionally, bars are given in colors.

Round 2

Reviewer 1 Report

The work does not show improvements compared to the previous version, no convincing answers have been given to the criticisms posed. The docking of compound 3 was not done at all, but the assay that would have highlighted its inhibitory potential on FLT3 was also not presented. In this latter respect, only one value has been added in the text, which the reviewer in this form cannot evaluate as there are no controls and it would be necessary for the authors to insert an ad hoc figure, both for docking and for the inhibitory activity on FLT3. 

Author Response

Dear Reviewer,

We are very sorry that in your opinion the manuscript did not improve, even though we incorporated all of your minor points and one of the two major points. With regard to the mentioned docking studies, we must say that this is a different research field which we do not conduct. However, with the results of the in vitro assays these docking studies are negligible as we demonstrated FLT3 inhibition of our compounds. With regard to the assay itself we are wondering about your comment as we measured not one but four concentrations of our compounds (50 nM /25 nM/ 12.5 nM/ 6.25 nM) and used gilteritinib, a known FLT3 inhibitor, as positive control. You will find these and more details in the Materials and Methods section (line 271 to 277). We only showed the lowest measured concentration in the Results part, as this was the most interesting finding. We did not yet provide IC50 values as we need to further titrate our compounds to lower concentrations. For that we ordered additional assay kit, which will take, however, another month to arrive.

With regard to the present study, we were able to deliver the proof of our theory because in the four measured concentrations we observed strong inhibition and could thus provide a rational for the observed impact of our compounds on AML cells. This proof as well as our efforts (establishing a complete new assay and subsequent testing of our compounds) was appreciated by the other reviewers. We think that with all the phytochemical work (all compounds were isolated and their structures elucidated for this study and not taken from an in house library), proliferation assays and studies on apoptotic effects of our compounds, it is justified to now include only preliminary results on the FLT3 inhibition into this manuscript. Even more so, as above all the fact that the compounds show inhibition was needed as a proof or our hypothesis.

We would be happy if you could reconsider your decision by taking all of these points into account.

Kind regards

Reviewer 2 Report

Thank you very much for all the explanations and the changes made. Thanks especially for the isolation chart in Supporting Information (SI), which makes it easy to understand the whole process.

In SI, please correct all "MeOD" and "MeOH" in compound (9) to "MeOH-d4" and all "DMSO" to "DMSO-d6" as they are written in the main manuscript (lines 216, 217). Please also correct SI "1H" and "13C" to "1H" and "13C" respectively - mentioned in the main manuscript in line 231. Please also include the calculated weights and those from the MS of all compounds.

In addition, in the future, I would encourage antitumor research to be also carried out on healthy cell lines, which would enable the assessment of the actual effect of compounds on the healthy cells tested and, possibly, the calculation of Selectivity Indexes. Additionally, I encourage you to extend your research to other cancer cell lines.

Author Response

Dear Reviewer,

Thank you very much for your nice comments and for appreciating our efforts. As suggested we corrected all points raised by you in the supporting information and included MS data of our compounds.

Thank you as well for your suggestions for future research aims. We definitely want to extend our research to additional cell lines and will also include healthy cell lines in our future studies.

Thank you once more for helping to improve our manuscript!

Kind regards

Reviewer 3 Report

The revised version addresses the comments raised by the reviewers on the original manuscript.

Author Response

Dear Reviewer,

Thank you very much for appreciating our efforts.

Kind regards

Reviewer 4 Report

The authors greatly improved their manuscript. They didn’t test their compound as FLT3 inhibitor but their arguments make sense and I agree with their conclusion. Accordingly, the manuscript can be publish as it stands now.

Author Response

Dear Reviewer,

Thank you very much for appreciating our efforts and for your constructive suggestions, which helped us to improve our manuscript.

Kind regards

Round 3

Reviewer 1 Report

The work in this form is merely descriptive of a generic cytotoxicity of molecules isolated from a plant matrix and would be more suitable for a phytotherapy or pharmacology journal. In order to merit publication in an authoritative "molecular" journal such as IJMS (ranking Q1: Biochemistry and Molecular Biology), it must at least be improved, and the way that this reviewer had suggested was to demonstrate that compound 1 and 3, among the many fractionated, has a molecular target, i.e. that compound 3 is an inhibitor of FLT3. It is appreciable that the authors have taken over the minor revisions suggested by this reviewer, which means that they have deemed them right by making them their own. But it is not enough to change the reviewer's opinion, as it would be necessary to demonstrate what was required in the major revisions, i.e. that (1) compound 3 has all the essential structural requirements to stably bind FLT3, through molecular docking - on this authors have not provided any direct evidence on their compound, but only generic circumstantial elements; in addition to demonstrating that (2) Compound 3 is really able to modify the biological activity of FLT3, through the FLT3 activity assay - but the data are preliminary, incomplete and therefore controversial following further investigation. This reviewer is not content with reading in lines 353-356 “Therefore, we investigated both compounds on their potential to reduce FLT3 kinase activity. Both compounds were effectively inhibiting FLT3 kinase at concentrations of 25 to 6.25 nM, with inhibition rates of 81% (1) and 88% (3) of 20 ng protein at the lowest concentration (DATA NOT SHOWN). " Data not shown has been deliberately indicated in small caps because we expected to see them. This reviewer expects the data to be shown in an ad hoc figure, in complete form, with the necessary checks and standard deviations. Since these results are essential for the work, they cannot be presented as preliminary data.